# A New Approach to Non-Invasive Microcirculation Monitoring: Quantifying Capillary Refill Time Using Oximetric Pulse Waves

**DOI:** 10.3390/s25020330

**Published:** 2025-01-08

**Authors:** Yuxiang Xia, Xinrui Wang, Zhe Guo, Xuesong Wang, Zhong Wang

**Affiliations:** 1School of Clinical Medicine, Tsinghua University, 30 Shuangqing Road, Haidian District Beijing, Beijing 102218, China; xiayx22@mails.tsinghua.edu.cn (Y.X.); xr-wang23@mails.tsinghua.edu.cn (X.W.); 2Beijing Tsinghua Changgung Hospital, School of Clinical Medicine, Tsinghua University, 168 Litang Road, Changping District, Beijing 102218, China; gza01482@btch.edu.cn

**Keywords:** microcirculation, capillary refill time, oxygen pulse wave, sepsis, non-invasive monitoring

## Abstract

(1) Background: To develop a novel capillary refill time measurement system and evaluate its reliability and reproducibility. (2) Methods: Firstly, the utilization of electromagnetic pressure technology facilitates the automatic compression and instantaneous release of the finger. Secondly, the employment of pressure sensing technology and photoelectric volumetric pulse wave analysis technology enables the dynamic monitoring of blood flow in distal tissues. Thirdly, the subjects were recruited to compare the average measurement time and the number of measurements required for successful measurements. The satisfaction of doctors and patients with the instrument was investigated through the administration of questionnaires. Finally, 71 subjects were recruited and divided into two groups, A and B. Three doctors repeated the measurement of the right index fingers of the subjects. In Group A, the same measuring instrument was used, and the consistency of the measurements was evaluated using the intragroup correlation coefficient. In Group B, one doctor repeated the measurement of each subject three times using the same measuring instrument, and the reproducibility of the CRT was evaluated using the analysis of variance of the repeated measurement data. (3) Results: The development of the capillary refill time meter was successful, with an average measurement time of 18 s and a single measurement. This study found that doctor–patient satisfaction levels were 98.3% and 100%, respectively. The intraclass correlation coefficient was 0.995 in Group A, and the *p*-value was greater than 0.05 in Group B. (4) Conclusions: The non-invasive monitoring of microcirculation has been rendered both rapid and effective, thus paving the way for the further mechanization and standardization of this process. The CRT, when measured using the capillary refill time meter test machine, demonstrated consistent and reproducible results, both when assessed by different researchers and when evaluated across varying measurement sets.

## 1. Introduction

In recent years, the development of hemodynamic theory and the advancement of clinical monitoring technologies have brought macrocirculation and microcirculation decoupling to the forefront of clinical research. For many critically ill patients, stabilizing macrocirculation may only represent the initial phase of treatment, with improvements in microcirculation proving pivotal to recovery [1,2,3,4]. The assessment of microcirculatory status in critically ill patients is predominantly achieved by measuring blood lactate, central venous oxygen saturation (ScvO2), the central venous-to-arterial carbon dioxide gap (CO2-gap), and other respiratory oxygen metabolism parameters; however, these invasive indicators are easily affected by respiratory function abnormalities, metabolic function abnormalities, measurement errors, and other factors. The need for the intuitive and real-time monitoring of microcirculation status has increased, but the challenge lies in directly monitoring the microcirculation status of the patient’s deep organs in practice. Consequently, there is an urgent need for the development of intuitive and accurate peripheral circulation monitoring techniques to enhance the current state of tissue perfusion monitoring in critically ill patients.

Capillary refill time (CRT) has been identified as the most intuitive monitoring index of peripheral circulatory blood flow status. It has been widely adopted in various clinical contexts, including patient condition assessment, treatment guidance, and critical care patient prognosis evaluation. The advantages of CRT include its intuitive nature, convenience, and non-invasiveness [5,6,7,8,9,10,11,12,13]. Patients resuscitated with the restoration of CRT as the resuscitation goal have been shown to exhibit reduced organ dysfunction at 72 h, decreased intravenous rehydration requirements, and lower 28-day mortality rates [11]. The International Sepsis Guidelines, the American Academy of Pediatrics, the World Health Organization, and the American Heart Association all currently recommend the incorporation of CRT measurements as a significant component of systemic assessment [14,15,16,17]. CRT can accurately reflect the microcirculatory status of the organism; however, its clinical application has been hindered by the need for human intervention in measuring parameters and the variability of measurement methods [18,19,20]. To obtain accurate and reliable quantitative data, the first fully automated CRT tabletop meter appeared in 2018 [21], which quantifies the CRT and reduces the interference of human factors. However, this instrument was not adopted into clinical practice due to several limitations, including high measurement error, low measurement success rate, extended measurement time, and substantial size. A survey of the current literature reveals that, as of yet, no CRT measurement device is being used in clinical practice. In light of this finding, the objective is to mechanize and standardize CRT measurement and combine it with artificial intelligence technology to obtain accurate quantitative data as the basis for clinical diagnosis and treatment. Conversely, we will take into full consideration the clinical use scenarios and develop instruments that can be truly applied to clinical use.

## 2. Materials and Methods

### 2.1. Detection Principles

Capillary refill time is measured artificially by pressing the peripheral skin tissues of the body so that local blood flow is interrupted; blood is squeezed to the periphery, skin discoloration occurs, local blood flow is gradually restored after the removal of pressure, and the time taken for the nail beds or the skin to become flushed red again is recorded. Therefore, the automatic measurement of capillary refill time requires automatic compression, the instantaneous release of peripheral skin, and the continuous monitoring of changes in distal tissue blood flow during this process.

According to the Lambert–Beer law [22], when the ventral surface of the distal phalanx of the right index finger is vertically irradiated using continuous parallel single-wavelength NIR light, the amount of absorbed light is directly proportional to the thickness of the irradiated tissue and the concentration of hemoglobin (oxyhemoglobin + reduced hemoglobin). As shown in Equation (1), when the amount of incident light, *I*_0_, is constant, the tissue thickness, *B*, is constant, and the molar absorption coefficients of hemoglobin for 900 nm near-infrared light, *K_HbO_*_2_ and *K_Hb_*, are constant, then the amount of transmitted light received, *I*, can reflect the change in the concentration of hemoglobin (oxygenated hemoglobin + reduced hemoglobin) in the blood flow, *C_HbO_*_2_ and *C_Hb_*. When the blood flow at the end of the finger changes, the *C_HbO_*_2_ and *C_Hb_* in the capillary bed will also change, and the transmitted light quantity, *I*, will also change in synchrony with the blood flow at the end of the finger.(1)$$A=\mathrm{lg}\left(\frac{I_{0}}{I}\right)=\left(K_{Hbo2}C_{Hbo2}+K_{Hb}C_{Hb}\right)\times B$$

It should be noted that *A* represents the absorbance, *I*_0_ represents the incident light intensity, *I* represents the transmitted light intensity, *K* represents the molar absorption coefficient, *B* represents the thickness of the absorbing layer, and *C* represents the concentration of light-absorbing substances. When the monochromatic light of a certain wavelength is irradiated into a certain medium, the degree of light being absorbed in the medium is related to the molar absorption coefficient of the light-blocking medium, the thickness of the light-blocking medium, and the concentration of the medium.

On this basis, we simultaneously designed an electromagnet pressurization device to achieve pressurization and the instantaneous release of the finger end. When the end of the finger is pressurized, the tissue is squeezed, arterial blood flow stops, the beat wave signal disappears, venous and capillary blood flow decreases, hemoglobin (oxyhemoglobin + reduced hemoglobin) concentration decreases, the tissue thickness is relatively reduced, the amount of absorbed light decreases, the amount of transmitted light increases, and the non-beat signal detected by the sensor is enhanced. When the pressurizing device releases the pressure, the tissue thickness increases, the concentration of hemoglobin (oxygenated hemoglobin + reduced hemoglobin) gradually recovers, the amount of absorbed light increases, the amount of transmitted light decreases, the non-pulsating pulsatile wave signals detected by the sensor gradually fall back, and pulsatile wave signals gradually recover with the restoration of the arterial blood flow; we consider that the refilling of the peripheral vasculature has been achieved when the transmissible photoelectric signals detected by the sensor recover to the baseline level before the pressurizing device (Figure 1).

### 2.2. Design and Implementation of Instrumented Measurement of Capillary Refill Time

Figure 2 shows a schematic of a mechanized CRT measurement in which a press actuator acts on the finger to affect the blood content in the terminal capillary bed, resulting in a change in blood oxygen concentration (Figure 2A). The blood oxygen concentration is expressed via a PPG, which is flipped to form a discrete time series (Figure 2B), with each element of the sequence consisting of the sampling time, *t_i_*, and the corresponding blood oxygen value, *x*(*t_i_*): $CRT=\left\{\left(\left.t_{i},x\left(t_{i}\right)\right|i=1,2\ldots N\right)\right\}$. The current instrument samples the optoelectronic signal at a frequency of *f* = 50 Hz, and usually, a complete CRT test consists of five phases, from Δ*t*_0_ to Δ*t*_4,_ with an overall time of about 15 s.

As illustrated in Figure 3, the internal structure diagram of the capillary refilling time measurement device comprises the following components: (1) The first is a blood oxygen and heart rate detection module that is suitable for detecting the blood oxygen and heart rate of the fingertip of the subject to be tested. (2) The second is a pressing structure that is spaced apart from the blood oxygen and heart rate detection module. The pressing structure is suitable for pressing the subject’s finger to be tested and is placed between the blood oxygen and heart rate detection module and the pressing structure. (3) The third is an elastic member connected to the pressing structure. The elastic member is suitable for applying a force away from the blood oxygen and heart rate detection module to the pressing structure. The apparatus comprises a magnetic suction member (4) that is connected to the pressing structure. An electromagnet (5) is connected to the magnetic suction member via a magnetic connection when the electromagnet is powered. A driving component (6) is connected to the electromagnet. The driving component is designed to move the electromagnet and the magnetic suction member, thereby driving the pressing structure to move closer to the subject’s finger to be tested, with the intention of pressing the subject’s finger to be tested.

The capillary refill time measuring instrument is mainly composed of a PPG sensor module, a pressure sensor, a power actuator module, a lithium battery pack, a microcontroller and its control circuit, an emergency release device, a scanning port for the wristband, and a high-definition display (Figure 4). Compared with the existing measuring instruments, this instrument has the following advantages: a compact size that is easy to use in emergency departments and outpatient wards; a unique electromagnetic design allowing the pressure device to instantly rebound and ensuring that the measurement results are true and reliable; the addition of a wristband scanning port, which enables scanning the patient’s wristband to automatically import the information of the test subject and storage of the test results and is easy to use in the clinic; a uniquely designed emergency release device allowing the pressure to be released and the measurement to be stopped at any time, which ensures the reliability and reliability of the measurement results and ensures the safety of the instrument; a visualization window design allowing the measurer and the subject to intuitively see the finger placement position and perform timely adjustments to improve the success rate of the measurement; and an ergonomic design to fit the hand shape so that the hand is relaxed and comfortable to avoid measurement errors caused by the stiffness of the hand.

We set up the measurement control logic for the instrumentation of capillary refill time based on the clinical application scenario and set the pressure of the machine on the finger to be automatically controlled by the software, stopping the pressure when the detected waveform of the electrical signal is a straight line and releasing the pressure automatically after it has been maintained for 5 s, which is equivalent to the pressing time of the finger using a final pressure of 5 s. It is important to note that a stable pulse waveform needs to be obtained before the start of the measurement to determine the baseline position. Data recording began with the recording of a stable pulse wave pattern 5 s before compression and continued until 10 s after the end of compression to ensure that the recorded blood flow refill curve returned to the baseline position. In addition, considering the safety of clinical use, protection control is set up so that the power supply can be cut off at any time during the whole measurement process, compression can be stopped at any time during compression through the ‘emergency release device’, and compression pressure can be released at any time, according to the clinical use scenarios set up in the control logic diagram shown in Figure 5.

### 2.3. Design and Realization of an Automated Processing System for Blood Oxygen Measurement Data

#### 2.3.1. Removal of Mutant Noise

In the process of analyzing and processing the blood oxygen measurement data, we found that under the influence of the photoelectric sensor, there is often a sudden change in the data (as shown in Figure 6), which appears randomly and has no fixed frequency; when it appears, it completely masks the original signal value, which belongs to the other noises in the fourth type mentioned above, and it is impossible to remove it by using traditional filtering methods. At the same time, this type of noise can cause serious interference to the signal calculation results, such as causing the peaks and valleys detection algorithm to fail, which, in turn, leads to measurement failure. Define Pa, Pb, Pc, and Pd to denote the four stages of the mutation noise, the beginning of the mutation, the mutation recovery, and the end of the mutation, respectively, as well as the corresponding instantaneous slopes of the four points as Ka, Kb, Kc, and Kd, respectively. Among these, if downward mutation noise occurs, the corresponding instantaneous slopes of the signals will, in general, reflect the characteristic of ‘down (Ka-Kb)-up (Kb-Kc)-down (Kc-Kd)’ in the lower right part of Figure 7A; if upward mutation noise occurs, the corresponding instantaneous slopes will reflect the characteristic of ‘up-down-up (Kc-Kd)’ in the lower left part of Figure 7A; and if upward mutation noise occurs, the corresponding instantaneous slopes will reflect the characteristic of ‘up-down-down-up (Kc-Kd)’ in the lower left part of Figure 7A. When a continuous mutation in time and long-duration mutation occurs, more than one mutation point will appear, at which time the mutation point Pb becomes a collection with (Pb1, Pb2, and Pb3) and so on appearing in it.

According to the different directions in the amplitude when it occurs, the noise can be classified into upward mutation and downward mutation, and this kind of noise can be further classified into three kinds according to the continuity in time: a single occurrence, a number of consecutive occurrences, and a long period of time continually occurring (shown in Figure 6). After a mutation noise, if the first-order difference between *Kb*2 and *Kb*3 set a point equivalent to the level of Ka, i.e., the first and last of the two single-mutation first-order difference features ‘down-up-down’, the situation in which the mutation noise occurs and is not recovered for a continuous period of time is called long-lasting mutation noise, as shown on the right side of Figure 7. According to the statistical analysis, when this situation occurs, the signal value will remain unchanged at the mutation position, i.e., Pb1 = Pb2 = Pb3, etc., and therefore, the corresponding first-order difference value also remains unchanged and equal to zero, i.e., *Kb*2 = *Kb*3 = 0.

Taking full consideration of the characteristics of the noise itself, we propose the idea of utilizing four-point slope features to locate the noise and remove it. The whole noise process is divided into four processes, start, mutation, mutation recovery, and end, and the key points of these four processes are identified in the signal by first-order difference. The instantaneous slope values were approximated by calculating the first-order difference of the oximetry data using the formula as shown in Equation (2) below.(2)$$x^{\prime }\left(t_{i}\right)=\frac{x\left(t_{i}\right)-x\left(t_{i-1}\right)}{t_{i}-t_{i-1}},i=2,3,\ldots ,N$$

According to the described time domain morphological features, the algorithm is designed to determine whether a noise point is suspicious by using the magnitude and direction of the change in the instantaneous slope. If the instantaneous slopes, $x^{\prime }\left(t_{i}\right)$ and $x^{\prime }\left(t_{i-1}\right)$, corresponding to a point, t_i_, and its previous point, *t_i_*_−1_, satisfy the relationship described in Equation (3), the current signal point is considered suspicious. $\Delta x^{\prime }$ is a slope threshold, which can be set as an observation or as a variance of the instantaneous slope close to Pb for a period of time; the threshold can be used to regulate the sensitivity of the algorithm, and the larger the threshold, the less sensitive the algorithm is to small-magnitude mutant noise. Further, the suspicious noise point is denoted as Pb. Subsequently, a further search for related points Pa, Pc, and Pd is required, and if all the points are matched successfully, then Pb is recognized as mutant noise. The matching process needs to take into account both continuous and persistent situations to finalize the removal of the noise.(3)$$\left|x^{\prime }\left(t_{i}\right)-x^{\prime }\left(t_{i-1}\right)\right|>\Delta x^{\prime }\;and\;x^{\prime }\left(t_{i}\right)\times x^{\prime }\left(t_{i-1}\right)<0$$

#### 2.3.2. Calculation of Pressure Release Recognition Points

The pressure release point is the point when the pressure is maintained for a period of time and then suddenly released. Prior to this point in time, the hemoglobin level in the blood remains stable at a low level due to the continuous maintenance of pressure; after the pressure release point, the oxygen level in the blood rises rapidly due to the refilling of the capillaries and then returns to the baseline level, with the overall characteristic of ‘level-rapidly rising’. The overall characteristic of ‘level-rapid rise’ is that as long as the blood oxygen measurement data locate such a section of the characteristics, you can find the pressure release point. In practice, however, the fluctuation of the sensitivity of the device and the presence of mutation noise make the implementation of the feature recognition algorithm based on the raw data more complicated. The identification of the press-release point seems to be more concerned with the overall trend of the signal, so we propose the quadratic Exponentially Weighted Averages-K (EWAK) algorithm. Exponentially weighted averages (EWA) were first applied to the PPG data to remove some of the high frequency and mutation noise effects while flattening the signal, based on which the corresponding instantaneous slopes of the signals were calculated using Equation (4); finally, the exponentially weighted averages were used again to obtain the smoother *EWAK* curve as the approximate first-order derivative curve of the PPG. Among these, the calculation of the exponentially weighted average can effectively reduce the influence of high-frequency information on the signal and facilitate the observation of the overall trend. The EWA calculation formula is as follows:(4)$$EWA\left(t_{i}\right)=\beta \times EWA\left(t_{i-1}\right)+\left(1-\beta \right)\times x\left(t_{i}\right),\beta \in \left[0,\left.1\right)\right.$$

$EWA\left(t_{i}\right)$, as an estimate of moment $t_{i}$, which can be substituted for the actual observation at this moment and is approximately equal to the average of the actual observations in the past 1/$\left(1-\beta \right)$ moments, indicates the weight of the data in the past period, which is an adjustable parameter, where larger values indicate that more past data are used, and smaller values indicate that the data at the current moment have a greater weight. When β=0.9, $EWA\left(t_{i}\right)$ is approximately the average of the last 10 values. However, setting β to a value that is too large means that more past data are used, which will cause a significant signal hysteresis that may result in points being recognized later than when they really occurred.

The instantaneous slopes are computed to approximate the first-order derivatives, and their peak points serve as the basis for critical point identification. A segment of the PPG signal is represented using a straight line, and the computed EWAK curve is represented using a dashed line. An example of the comparison of the PPG signal with the corresponding quadratic sliding average-derivative values is shown in Figure 8. The EWAK curve corresponding to the oximetry data near the point of pressure release shows a steady characteristic, i.e., it rises rapidly after a nearly horizontal curve and reaches the maximum point of the whole curve, as shown in the figure using the red highlights. The EWAK curve approximates a first-order derivative, and the maximum point indicates the point of maximum velocity in the oximetry data, which corresponds to the fastest point in capillary filling in the real sense phase. Therefore, the algorithm starts from the maximum point and looks for a smooth curve to the left, and at the transition between the smooth curve and the rising curve, the corresponding point in the oximetry data is the point to be searched for the pressure release point, as shown by the intersection of the vertical dashed line and the PPG signal in the figure.

### 2.4. Evaluation of the Application Effect of This Instrument

In order to verify and evaluate the application effect of the instrument in a real medical environment, subjects were recruited in the Physical Examination Center of Tsinghua Changgeng Hospital in Beijing. The inclusion criteria are as follows: healthy adults ≥18 years old with no prior severe organ dysfunction or peripheral vascular disease. The exclusion criteria are as follows: diabetes mellitus; high blood pressure; systemic vasculitis; upper limb artery occlusion; pregnant or lactating women; gray nails, painted nail polish, or a nail bed that is too thick, preventing the machine from effectively measuring data; the absence of the right index finger for measurement; and did not sign the informed consent or did not agree to participate in the research. Each subject was measured with the existing benchtop measuring instrument and the proposed instrument, and the average measurement time and the number of successful measurements required for the two measurement methods were compared. During the measurement, we ensured that the ambient temperature was between 20 and 25 °C, the subject stayed in the environment for more than 30 min, the subject was seated, the fingertip was raised to the height of the heart, the measurement site was the right index finger, and the interval between two measurements was 5 min. Next, the instrument was tested in the Health Management Center, the Department of General Medicine, the Department of Endocrinology, and the Intensive Liver Care Unit of Tsinghua Changgeng Hospital in Beijing. A questionnaire survey on the satisfaction of all doctors and patients using this instrument was also used to collect data.

### 2.5. Reliability and Reproducibility of the Proposed Device for Assessing Capillary Refill Time

The inclusion criteria are as follows: healthy people and patients with type 2 diabetes in the Physical Examination Center of Tsinghua Changgeng Hospital, Beijing, aged ≥18 years old. The exclusion criteria are as follows: serious organ dysfunction; systemic vasculitis; upper limb artery occlusion; upper limb artery thrombosis; pregnant or lactating women; gray nails, painted nail polish, or a nail bed that is too thick, preventing the machine from effectively measuring data; the absence of the right index finger for measurement; and did not sign the informed consent or did not agree to participate in the research. Subjects were included according to the exclusion criteria and randomly divided into two groups: A and B.

First, three physicians trained in standard measurement methods used the instrument to measure the Group A subjects, and the intragroup correlation coefficient (ICC) was used to evaluate the consistency between the different measures. Then, another doctor used the instrument to measure the subjects in Group B, and the measurements were repeated 3 times for each subject in Group B to compare the repeatability of the measurement results. During the whole measurement process, we ensured that the ambient temperature was between 20 and 25 °C, the subject stayed in the environment for more than 30 min, and the fingertip was raised to roughly the same height as the heart and put into the detection window according to the prompts. The instrument automatically pressed down after obtaining a stable pulse wave pattern, stopped pressing when the detected electrical signal waveform was in a straight line, and automatically released the pressure after maintaining it for 5 s. We continued recording until 10 s after compression to ensure that the recorded blood refill curve returned to the baseline position. The interval between each measurement was 5 min.

In order to conduct a quantitative evaluation of the measurement tool, the minimum sample size required for the reliability and validity test needed to be calculated. The calculation formula is as follows: given the test level, *α* = 0.05 and *β* = 0.8, the correlation coefficient was expected to be 0.6, and the minimum sample size was 20. Epidata3.0 was used to establish a database, two people entered the data, and the data were imported into SPSS 25.0 statistical software. The measurement data were expressed as mean ± standard deviation, and the results were compared using ANOVA with repeated measurement data.(5)$$N={\left[\frac{Z_{\alpha /2}+Z_{\beta }}{0.5\ln\frac{1+r}{1-r}}\right]}^{2}+3$$

## 3. Results and Discussion

### 3.1. Testing of Automated Systems for Processing Oximetry Data

In order to evaluate the denoising effect of the algorithm on mutant noise, a common metric for measuring the difference between values is used, namely, the root mean square error (RMSE), which is commonly used in the field of signal processing to evaluate the difference between the real signal without noise and the filtered and denoised signal, with smaller values indicating better filtering. The RMSE calculation formula is shown in (6). In the formula, N denotes the length of the signal, ${CRT}_{r}$ denotes the signal that does not contain mutation noise, and ${CRT}_{de}$ denotes the signal that has been processed by the denoising algorithm. The overall evaluation process was as follows: (1) the construction of ${CRT}_{r}$; (2) the construction of random mutation noise, *Noise*; (3) the addition of random mutation noise, *Noise*, to ${CRT}_{r}$ to obtain the oximetry data, ${CRT}_{sim}$; (4) the processing of ${CRT}_{sim}$ using the mutation noise removal algorithm to obtain ${CRT}_{de}$; and (5) the evaluation of the effectiveness of the algorithm using the RMSE for ${CRT}_{r}$ and ${CRT}_{de}$.(6)$$RMSE=\sqrt{\frac{1}{N}\left[\sum_{k=1}^{N}{\left({CRT}_{r}\left(k\right)-{CRT}_{de}\left(k\right)\right)}^{2}\right]}$$

The single oximetry data processed by this system are in the form of a discrete time series, where each element of the series consists of the sampling time, $t_{i}$, and the corresponding PPG value, $CRT=\left\{\left(\left.t_{i},x\left(t_{i}\right)\right|i=1,2\ldots N\right)\right\}$. Using a CRT test with a total duration of T=15 s and a sampling frequency of F=50 Hz, we can obtain N=T×F=750; the number of mutations in Noise is 34. Using this algorithm for 40 manually created ${CRT}_{r}$ and to calculate the average RMSE, the minimum RMSE = 119.3 is obtained when the slope threshold is set as follows: $\Delta x^{\prime }=100$. Applied to oximetry data, an example of the algorithm denoising effect is shown in Figure 8. Figure 9A was intercepted from a section of the CRT test process when the signal was continuously pressed; the significance of this section of noise elimination is that it can effectively improve the accuracy of the subsequent keypoint identification algorithm because the algorithm needs to calculate the average variance of the signal over a period of time, and if too much mutation noise exists, it will have a serious impact on the average variance. Figure 9B was intercepted from a segment of the signal at baseline during the CRT test when the signal regularly produced variations in response to heartbeats. The significance of this noise removal is that it effectively improves the accuracy of the subsequent peak point identification algorithm, as the algorithm relies on the identification of poles within a region, which can easily overwrite the poles in the normal signal if an abrupt change occurs.

### 3.2. Evaluation of the Application Effect of This Instrument

This study recruited 50 healthy adult subjects who met the inclusion criteria for physical examination at the Physical Examination Center of Tsinghua Changgeng Hospital in Beijing in December 2024. This study compared the average measurement time and the number of successful measurements required by two different measurement methods. The results indicated that the benchtop measuring instrument had an average measurement time of 42 s and a higher incidence of measurement failures, necessitating 3–5 repeated attempts to achieve success. In contrast, the alternative instrument achieved an average measurement time of 18 s and required only one attempt for successful measurement. Additionally, clinical testing was conducted from October to November 2024 at the Health Management Center, Department of General Medicine, Department of Endocrinology, and Intensive Liver Care Unit of Tsinghua Changgeng Hospital in Beijing—a questionnaire collected satisfaction data from 58 doctors and 142 patients. The results are summarized in Table 1. For the benchtop measuring instrument, the doctor satisfaction rate was 17.2%, with dissatisfaction primarily attributed to low measurement success rates, prolonged measurement times, the need for a power connection during use, and its large, bulky size. Patient satisfaction was 50.7%, with complaints focusing on long measurement times, loud instrument noise, and finger pain. Conversely, the alternative instrument achieved 100% patient satisfaction and 98.3% doctor satisfaction, with only one endocrinologist expressing dissatisfaction due to the instrument’s limitation in measuring finger CRT. Considering the potential value of CRT measurement for diabetic feet, it is suggested that an instrument capable of measuring foot CRT be developed. Future studies could explore new versions tailored for specific applications, such as a device dedicated to diabetic foot assessment.

### 3.3. Reliability and Reproducibility of the Proposed Device to Assess Capillary Refill Time

A total of 71 subjects were included in this study, with 31 assigned to Group A (ICC = 0.995), which demonstrates excellent inter-rater reliability across different measurements. For the 40 subjects in Group B, the results of variance analysis for repeated measurement data are presented in Table 2. The findings indicate that there was no statistically significant difference among the three measurements (*p* > 0.05), suggesting that the device exhibits good repeatability in assessing capillary refill time.

## 4. Conclusions

When circulatory dysfunction occurs in the organism, peripheral circulation is often sacrificed first to ensure the blood perfusion of important organs; when circulatory function is gradually restored, it is restored last. Therefore, peripheral circulation monitoring indices can sensitively reflect the status of systemic microcirculation blood flow. Based on the working mechanism of peripheral oxygen saturation monitoring technology commonly used in clinical practice, we searched for a method to continuously monitor fingertip blood flow, proposed a basic design for instrumentation to measure the indices related to capillary refill time, and improved the existing equipment to address deficiencies in the control of compression time and force, the mechanism of determining the individualized capillary closure time, and the design of the compression actuating device. This system can instrument and standardize the measurement of the parameters related to capillary refill time, minimize measurement error, and improve the reliability and accuracy of the clinical index of capillary refill time; the measurement process is also non-invasive and simple. Preliminary clinical studies have shown that the measurement is stable and has good practical value and application prospects, providing a new method for peripheral circulation monitoring.

## 5. Patents

We have applied for three invention patents, all of which have been disclosed and are currently under substantive examination. (Application Nos. 202410514816X, 2024106642551, and 2024107379296).

## Figures and Tables

**Figure 1 sensors-25-00330-f001:**
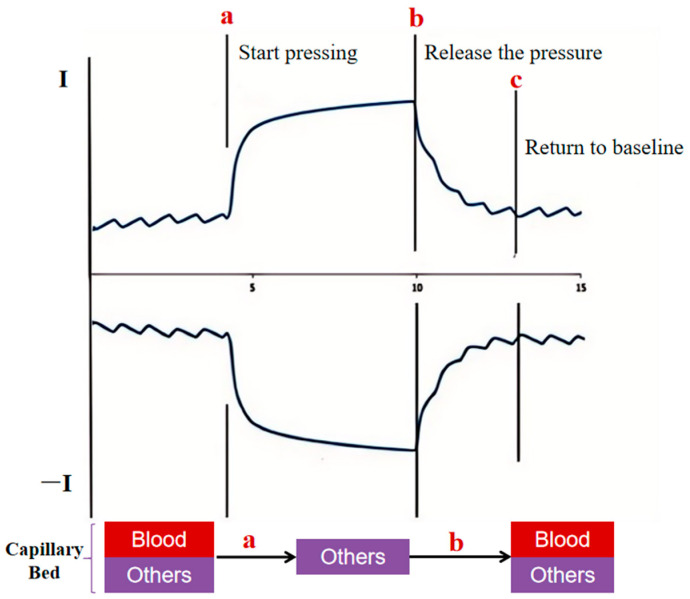
Schematic diagram of instrumented measurement of capillary refill time. a: start pressing; b: release the pressure; c: return to baseline. *I* represents the transmitted light intensity.

**Figure 2 sensors-25-00330-f002:**
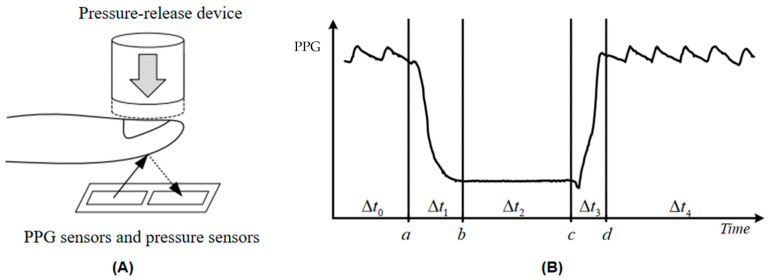
(**A**) Electromagnet-based press-release device capable of instant rebound after power failure that does not interfere with blood return like other mechanical compression devices. PPG sensors are able to continuously monitor the amount of light reflected from the end of the finger, and pressure sensors monitor the amount of pressure in real time. (**B**) PPG signal curve. The regular undulation with the heartbeat is called the baseline state. When the instrument automatically detects the baseline state for a period of time, the pressing device starts to apply pressure to the finger, and this period of time is recorded as ∆*t*_0_; with the gradual increase in the pressure, the arterial blood in the capillary bed is gradually drained, and the concentration of hemoglobin drops sharply, which is reflected in the figure as a section of the PPG signal curve that decreases rapidly. The PPG signal curve decreases rapidly when a certain pressure threshold value (3–5 N) is reached, the pressure device stops increasing the pressure, and this period of time is recorded as ∆*t*_1_. Since the end of the finger is essentially emptied of blood, it is reflected in the figure as a segment of the PPG signal curve that is essentially free of fluctuations, and the period of time during which the signal remains steady at a low level is denoted as ∆*t*_2_. The electromagnet is released instantaneously, the blood in the capillaries at the end of the finger gradually fills up, and the blood circulation and hemoglobin concentration will return to pre-pressure levels for a period of time, denoted as ∆*t*_3_. After the PPG signal curve returns to a stable fluctuation, it then continues to be observed for a period of time as a sign of a successful test, which is denoted as ∆*t*_4_. a: initiation of compression; b: maintenance of compression; c: release of pressure; d: return of blood to return to baseline status.

**Figure 3 sensors-25-00330-f003:**
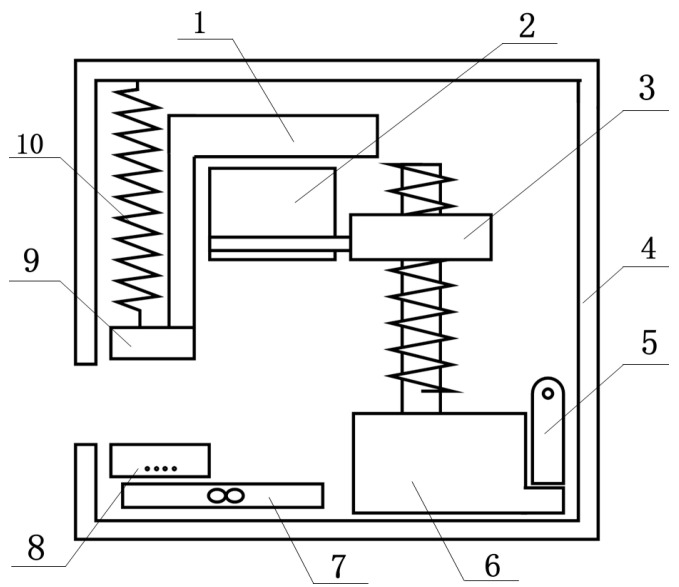
Diagram illustrating the internal structure of the capillary refill time measuring instrument. The instrument comprises the following components: (1) a magnetic suction plate; (2) an electromagnet; (3) a linkage mechanism; (4) a housing; (5) a mechanical emergency release mechanism; (6) a direct current motor; (7) a pressure sensor; (8) a blood oxygen- and heart rate-detecting module; (9) a pressing structure; and (10) a reset spring. Patient information can be entered in an optional manner by scanning the wristband or through direct entry. Following the placement of the finger within the detection area, i.e., between the blood oxygen and heart rate detection module 8 and the pressing structure 9, the system is ready for operation. The activation of the detection is initiated through the microcontroller control of the DC motor action: rotating screw 11, driving slider 3 down, electromagnet 2 driving the magnetic suction member 1 down, and driving the pressing structure 9 down, resulting in the finger capillary blood being emptied. Concurrently, the real-time pressure is detected, and when it reaches 3 N, the pressing action is halted. Following a period of maintenance (5–7 s), the electromagnet 2 is deactivated, ceasing its magnetic attraction to the magnetic suction member 1. Concurrently, the elasticity member 10 rapidly releases the pressing structure 9, lifting the pressing action and initiating the refilling of the finger capillaries with blood. During this period, the measurement process is completed by the recording of data changes in heart rate and blood oxygen. That is to say, the time from blood emptying to refilling by the blood oxygen heart rate detection module 8 is measured and recorded. The capillary refilling time is then analyzed and calculated in order to obtain the capillary refilling time.

**Figure 4 sensors-25-00330-f004:**
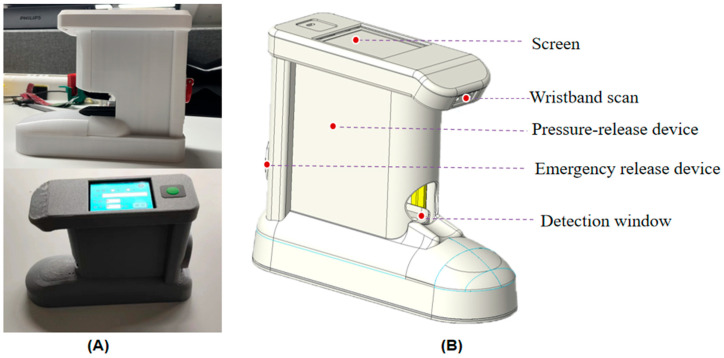
(**A**) Photograph of the portable capillary refill time meter; (**B**) 3D view of the capillary refill time meter.

**Figure 5 sensors-25-00330-f005:**
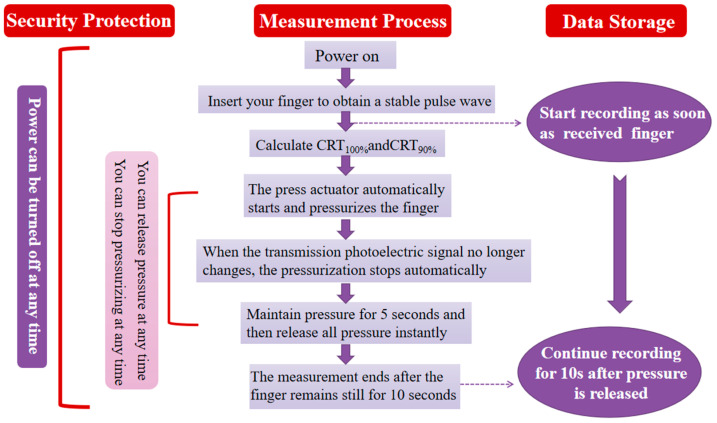
Control logic diagram of capillary refill time measuring instrument tester.

**Figure 6 sensors-25-00330-f006:**
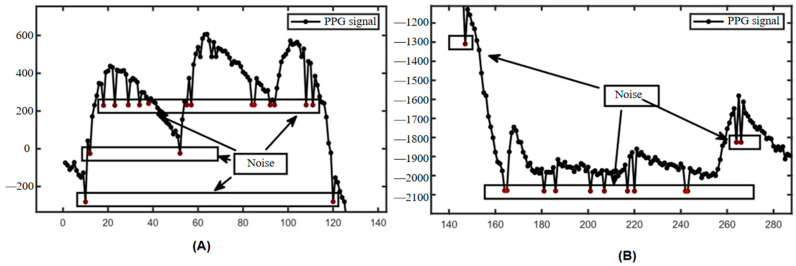
Schematic diagram of mutation noise. In addition to the photoelectric volume pulse wave signal, the noise collected during signal acquisition is influenced by multiple factors including sensor sensitivity, contact tightness, skin hydration level, and epidermal thickness at the measurement site. (**A**,**B**) are noise diagrams. This type of noise manifests randomly without a fixed frequency. Upon occurrence, it completely obscures the original signal and cannot be eliminated through conventional filtering techniques. Furthermore, this noise significantly interferes with signal processing, leading to failures in algorithms designed for peak and valley point detection, ultimately resulting in measurement inaccuracies.

**Figure 7 sensors-25-00330-f007:**
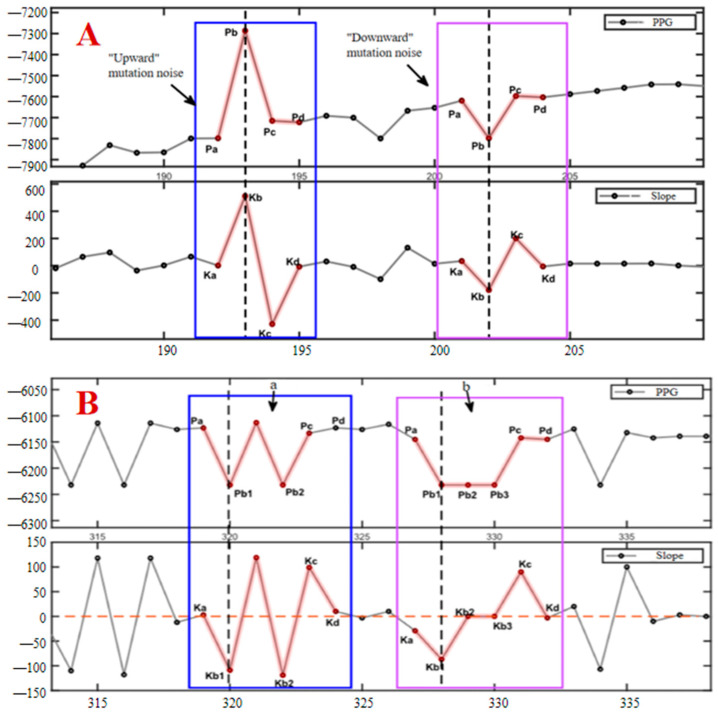
Classification of mutation noise. The upper vertical coordinate is the oximetry data (PPG), and the lower graph shows the corresponding instantaneous slope of the signal. (**A**) shows a schematic diagram of ‘upward’ and ‘downward’ mutation noise, and (**B**) shows a schematic diagram of continuous and sustained mutation noise, where a denotes continuous mutation noise and b denotes sustained mutation noise.

**Figure 8 sensors-25-00330-f008:**
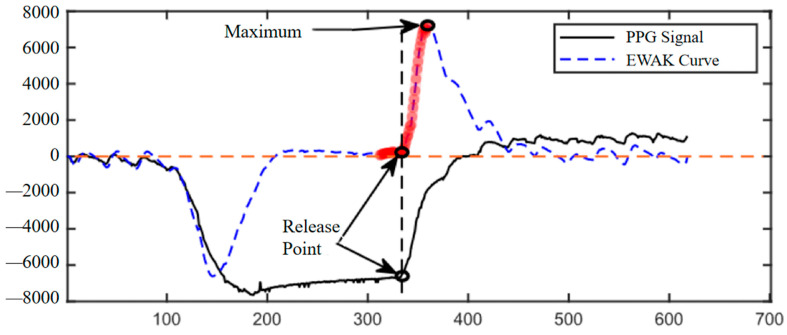
CRT data and their EWAK curve.

**Figure 9 sensors-25-00330-f009:**
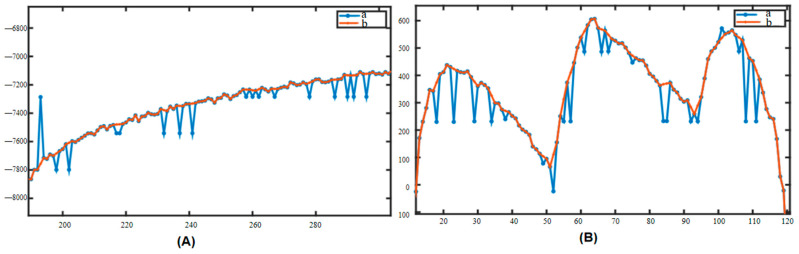
Example of denoising effect of this algorithm. In the figure, a represents the initial value of CRT, and b is the CRT after removing the noise. (**A**): The signal anomalies resulting from continuous pressing during a CRT test have been filtered out. This noise reduction is crucial as it significantly enhances the accuracy of the subsequent key point identification algorithm. The algorithm requires calculating the mean variance of the signal over a specified period, and excessive transient noise can severely affect this calculation; (**B**): The signal originates from the baseline of a CRT test, where it fluctuates in accordance with the rhythmic pattern of heartbeats.

**Table 1 sensors-25-00330-t001:** Comparison of the two measurement methods.

Items	Bench Measuring Instrument	This Instrument
Average measurement time (seconds)	42	18
Number of successful measurements required (times)	3–5	1
Doctor satisfaction (%)	17.2% (10/58)	98.3% (57/58)
Patient satisfaction (%)	50.7% (72/142)	100% (142/142)

**Table 2 sensors-25-00330-t002:** Results of variance analysis of repeated measurement data.

Sources of Variation		SS	V	MS	F	*p*
Healthy subjects	Treatment (intercolumn)	0.004750	2	0.002375	1.693	1.2043
	inter-individual differences	15.68	19	0.8254	588.3	<0.0001
	residual	0.05332	38	0.001403		
	total	15.74	59			
Diabetic subject	Treatment (intercolumn)	0.02719	2	0.01360	0.7455	0.4077
	inter-individual differences	28.94	19	1.523	83.53	<0.0001
	residual	0.6929	38	0.01824		
	total	29.66	59			

## Data Availability

The datasets used and/or analyzed during the current study are available from the corresponding author upon reasonable request.

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
