# Peer review of "A New Approach to Non-Invasive Microcirculation Monitoring: Quantifying Capillary Refill Time Using Oximetric Pulse Waves"

_sensors, 2025, doi:10.3390/s25020330_

Round 1

Reviewer 1 Report

Comments and Suggestions for Authors

The paper presents a novel device to measure capillary refill time (CRT) using oxymetric pulse wave analysis. However, there are several limitations that need to be addressed. Firstly, the device was tested by only two individuals. This small sample size and limited number of testers significantly restricts the generalizability and reliability of the findings. Additionally, the study lacks comparison to other existing CRT measurement devices and the standard manual capillary refill test. Such comparisons are crucial to establishing the device's accuracy, consistency, and potential advantages over current methods.

In Table 1, the precision of CRT measurements is presented in milliseconds with two digits. In practice, CRT is usually measured in whole seconds due to the limitations of human perception and the variability in manual assessment. The precision of milliseconds may not translate into a meaningful clinical difference. What is the clinical significance and practical need for millisecond measurement?

Reviewer 2 Report

Comments and Suggestions for Authors

This article is very interesting in the field of studying blood filling of blood vessels. In the article, the authors proposed a new method for recording a pulse wave, which in my opinion is quite promising. The article describes in detail the structure of the device, the main aspects of recording and processing signals, as well as the analysis of the obtained data. The article fully complies with the topic of the journal.

To improve the manuscript, I strongly advise the authors to correct the following points:

1. Add quantitative metrics of the results to the abstract

2. Expand the analysis of the current state of the problem. At the moment, the introduction looks very poor.

3. Add a more extensive description of the recorder. Perhaps it is necessary to add a detailed structural diagram

4. Expand the description of formula (1)

I would especially like to note that the results were obtained on a very small sample (10 people). I believe that the authors should increase the sample size for more scientific and significant conclusions in future publications .
